# CRITICAL PERCOLATION AS A FRAMEWORK TO ANALYZE THE TRAINING OF DEEP NETWORKS

**Zohar Ringel**
Racah Institute of Physics
The Hebrew University of Jerusalem
zohar.ringel@mail.huji.ac.il.

**Rodrigo de Bem**[*]
Department of Engineering Science
University of Oxford
rodrigo@robots.ox.ac.uk

## ABSTRACT

In this paper we approach two relevant deep learning topics: i) tackling of graph structured input data and ii) a better understanding and analysis of deep networks and related learning algorithms. With this in mind we focus on the topological classification of reachability in a particular subset of planar graphs (Mazes). Doing so, we are able to model the topology of data while staying in Euclidean space, thus allowing its processing with standard CNN architectures. We suggest a suitable architecture for this problem and show that it can express a perfect solution to the classification task. The shape of the cost function around this solution is also derived and, remarkably, does not depend on the size of the maze in the large maze limit. Responsible for this behavior are rare events in the dataset which strongly regulate the shape of the cost function near this global minimum. We further identify an obstacle to learning in the form of poorly performing local minima in which the network chooses to ignore some of the inputs. We further support our claims with training experiments and numerical analysis of the cost function on networks with up to $128$ layers.

## 1 INTRODUCTION

Deep convolutional networks have achieved great success in the last years by presenting human and super-human performance on many machine learning problems such as image classification, speech recognition and natural language processing (LeCun et al. (2015)). Importantly, the data in these common tasks presents particular statistical properties and it normally rests on regular lattices (e.g. images) in Euclidean space (Bronstein et al. (2016)). Recently, more attention has been given to other highly relevant problems in which the input data belongs to non-Euclidean spaces. Such kind of data may present a graph structure when it represents, for instance, social networks, knowledge bases, brain activity, protein-interaction, 3D shapes and human body poses. Although some works found in the literature propose methods and network architectures specifically tailored to tackle graph-like input data (Bronstein et al. (2016); Bruna et al. (2013); Henaff et al. (2015); Li et al. (2015); Masci et al. (2015a;b)), in comparison with other topics in the field this one is still not vastly investigated.

Another recent focus of interest of the machine learning community is in the detailed analysis of the functioning of deep networks and related algorithms (Daniely et al. (2016); Ghahramani (2015)). The minimization of high dimensional non-convex loss function by means of stochastic gradient descent techniques is theoretically unlikely, however the successful practical achievements suggest the contrary. The hypothesis that very deep neural nets do not suffer from local minima (Dauphin et al. (2014)) is not completely proven (Swirszcz et al. (2016)). The already classical adversarial examples (Nguyen et al. (2015)), as well as new doubts about supposedly well understood questions, such as generalization (Zhang et al. (2016)), bring even more relevance to a better understanding of the methods.

In the present work we aim to advance simultaneously in the two directions described above. To accomplish this goal we focus on the topological classification of graphs (Perozzi et al. (2014);

---

[*]Rodrigo de Bem is also Assistant Professor at the Federal University of Rio Grande, Rio Grande, Brazil (rodrigobem@furg.br).

Scarselli et al. (2009)). However, we restrict our attention to a particular subset of planar graphs constrained by a regular lattice. The reason for that is threefold: i) doing so we still touch upon the issue of real world graph structured data, such as the 2D pose of a human body (Andriluka et al. (2014); Jain et al. (2016)) or road networks (Masucci et al. (2009); Viana et al. (2013)); ii) we maintain the data in Euclidean space, allowing its processing with standard CNN architectures; iii) this particular class of graphs has various non-trivial statistical properties derived from percolation theory and conformal field theories (Cardy (2001); Langlands et al. (1994); Smirnov & Werner (2001)), allowing us to analytically compute various properties of a deep CNN proposed by the authors to tackle the problem.

Specifically, we introduce Maze-testing, a specialized version of the reachability problem in graphs (Yu & Cheng (2010)). In Maze-testing, random mazes, defined as $L$ by $L$ binary images, are classified as solvable or unsolvable according to the existence of a path between given starting and ending points in the maze (vertices in the planar graph). Other recent works approach maze problems without framing them as graphs (Tamar et al. (2016); Oh et al. (2017); Silver et al. (2017)). However, to do so with mazes (and maps) is a common practice in graph theory (Biggs et al. (1976); Schrijver (2012)) and in applied areas, such as robotics (Elfes (1989); Choset & Nagatani (2001)). Our Maze-testing problem enjoys a high degree of analytical tractability, thereby allowing us to gain important theoretical insights regarding the learning process. We propose a deep network to tackle the problem that consists of $O(L^2)$ layers, alternating convolutional, sigmoid, and skip operations, followed at the end by a logistic regression function. We prove that such a network can express an exact solution to this problem which we call the optimal-BFS (breadth-first search) minimum. We derive the shape of the cost function around this minimum. Quite surprisingly, we find that gradients around the minimum do not scale with $L$. This peculiar effect is attributed to rare events in the data.

In addition, we shed light on a type of sub-optimal local minima in the cost function which we dub "neglect minima". Such minima occur when the network discards some important features of the data samples, and instead develops a sub-optimal strategy based on the remaining features. Minima similar in nature to the above optimal-BFS and neglect minima are shown to occur in numerical training and dominate the training dynamics. Despite the fact the Maze-testing is a toy problem, we believe that its fundamental properties can be observed in real problems, as is frequently the case in natural phenomena (Schmidt & Lipson (2009)), making the presented analytical analysis of broader relevance.

Additionally important, our framework also relates to neural network architectures with augmented memory, such as Neural Turing Machines (Graves et al. (2014)) and memory networks (Weston et al. (2014); Sukhbaatar et al. (2015)). The hot-spot images (Fig. 7), used to track the state of our graph search algorithm, may be seen as an external memory. Therefore, to observe how activations spread from the starting to the ending point in the hot-spot images, and to analyze errors and the landscape of the cost function (Sec. 5), is analogous to analyze how errors occur in the memory of the aforementioned architectures. This connection gets even stronger when such memory architectures are employed over graph structured data, to perform task such as natural language reasoning and graph search (Weston et al. (2015); Johnson (2017); Graves et al. (2016)). In these cases, it can be considered that their memories in fact encode graphs, as it happens in our framework. Thus, the present analysis may eventually help towards a better understanding of the cost functions of memory architectures, potentially leading to improvements of their weight initialization and optimization algorithms thereby facilitating training (Mishkin & Matas (2015)).

The paper is organized as follows: Sec. 2 describes in detail the Maze-testing problem. In Sec. 3 we suggest an appropriate architecture for the problem. In Sec. 4 we describe an optimal set of weights for the proposed architecture and prove that it solves the problem exactly. In Sec. 5 we report on training experiments and describe the observed training phenomena. In Sec. 6 we provide an analytical understanding of the observed training phenomena. Finally, we conclude with a discussion and an outlook.

## 2 MAZE-TESTING

Let us introduce the Maze-testing classification problem. Mazes are constructed as a random two dimensional, $L \times L$, black and white array of cells (I) where the probability ($\rho$) of having a black cell is given by $\rho_c = 0.59274(6)$, while the other cells are white. An additional image (H$_0$), called

the initial hot-spot image, is provided. It defines the starting point by being zero (*Off*) everywhere except on a $2 \times 2$ square of cells having the value 1 (*On*) chosen at a random position (see Fig.7). A Maze-testing sample (i.e. a maze and a hot-spot image) is labelled *Solvable* if the ending point, defined as a $2 \times 2$ square at the center of the maze, is reachable from the starting point (defined by the hot-spot image) by moving horizontally or vertically along black cells. The sample is labelled *Unsolvable* otherwise.

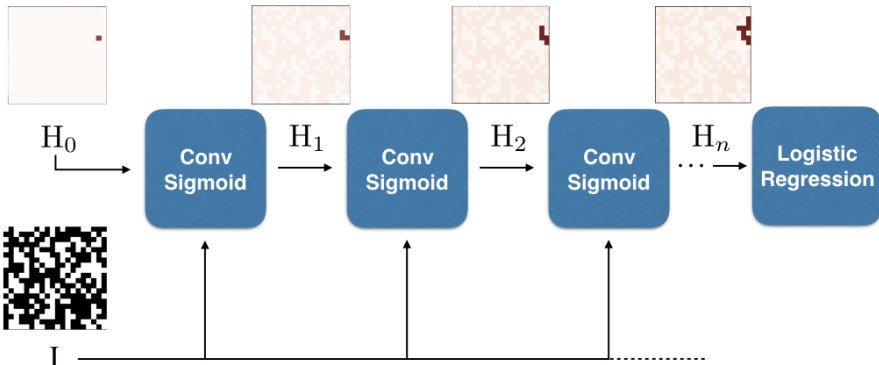

**Figure 1: Dataset, Architecture, and the Breadth-First Search optimum.** A maze-testing sample consists of a maze (I) and an initial hot-spot image ($H_0$). The proposed architecture processes $H_0$ by generating a series of hot-spot images ($H_{i>0}$) which are of the same dimension as $H_0$ however their pixels are not binary but rather take on values between 0 (*Off*, pale-orange) and 1 (*On*, red). This architecture can represent an optimal solution, wherein the red region in $H_0$ spreads on the black cluster in I to which it belongs. Once the spreading has exhausted, the Solvable/Unsolvable label is determined by the values of $H_n$ at center (ending point) of the maze. In the above example, the maze in question is Unsolvable, therefore the *On* cells do not reach the ending point at the center of $H_n$.

A maze in a Maze-testing sample has various non-trivial statistical properties which can be derived analytically based on results from percolation theory and conformal field theory (Cardy (2001); Langlands et al. (1994); Smirnov & Werner (2001)). Throughout this work we directly employ such statistical properties, however we refer the reader to the aforementioned references for further details and mathematical derivations.

At the particular value chosen for $\rho$, the problem is at the percolation-threshold which marks the phase transition between the two different connectivity properties of the maze: below $\rho_c$ the chance of having a solvable maze decays exponentially with $r$ (the geometrical distance between the ending and starting points). Above $\rho_c$ it tends to a constant at large $r$. Exactly at $\rho_c$ the chance of having a solvable maze decays as a power law ($1/r^\eta, \eta = 5/24$). We note in passing that although Maze-testing can be defined for any $\rho$, only the choice $\rho = \rho_c$ leads to a computational problem whose typical complexity increases with $L$.

Maze-testing datasets can be produced very easily by generating random arrays and then analyzing their connectivity properties using breadth-first search (BFS), whose worse case complexity is $O(L^2)$. Notably, as the system size grows larger, the chance of producing solvable mazes decay as $1/L^\eta$, and so, for very large $L$, the labels will be biased towards unsolvable. There are several ways to de-bias the dataset. One is to select an unbiased subset of it. Alternatively, one can gradually increase the size of the starting-point to a starting-square whose length scales as $L^\eta$. Unless stated otherwise, we simply leave the dataset biased but define a normalized test error ($err$), which is proportional to the average mislabeling rate of the dataset divided by the average probability of being solvable.

## 3  THE ARCHITECTURE

Here we introduce an image classification architecture to tackle the Maze-testing problem. We frame maze samples as a subclass of planar graphs, defined as regular lattices in the Euclidean space, which can be handle by regular CNNs. Our architecture can express an exact solution to the problem and, at least for small Mazes ($L \leq 16$), it can find quite good solutions during training. Although applicable to general cases, graph oriented architectures find it difficult to handle large sparse graphs due to

regularization issues (Henaff et al. (2015); Li et al. (2015)), whereas we show that our architecture can perform reasonably well in the planar subclass.

Our network, shown in Fig. (7), is a deep feedforward network with skip layers, followed by a logistic regression module. The deep part of the network consists of $n$ alternating convolutional and sigmoid layers. Each such layer $(i)$ receives two $L \times L$ images, one corresponding to the original maze (I) and the other is the output of the previous layer ($H_{i-1}$). It performs the operation $H_i = \sigma(K_{hot} * H_{i-1} + K * I + b)$, where $*$ denotes a $2D$ convolution, the $K$ convolutional kernel is $1 \times 1$, the $K_{hot}$ kernel is $3 \times 3$, $b$ is a bias, and $\sigma(x) = (1 + e^{-x})^{-1}$ is the sigmoid function. The logistic regression layer consists of two perceptrons $(j = 0, 1)$, acting on $H_n$ as $[p_0, p_1]^T = W_j \vec{H}_n + \vec{b}_{reg}$, where $\vec{H}_n$ is the rasterized/flattened version of $H_n$, $W_j$ is a $2 \times L^2$ matrix, and $\vec{b}_{reg}$ is a vector of dimension 2. The logistic regression module outputs the label *Solvable* if $p_1 \geq p_0$ and *Unsolvable* otherwise. The cost function we used during training was the standard negative log-likelihood.

# 4 AN OPTIMAL SOLUTION: THE BREADTH-FIRST SEARCH MINIMUM

As we next show, the architecture above can provide an exact solution to the problem by effectively forming a cellular automaton executing a breadth-first search (BFS). A choice of parameters which achieves this is $\lambda \geq \lambda_c = 9.727 \pm 0.001$, $K_{hot} = [[0, \lambda, 0], [\lambda, \lambda, \lambda], [0, \lambda, 0]]$, $K = 5\lambda_c$, $b = -5.5\lambda_c$, $[W]_{jq} = (-1)^j \lambda \delta_{q_{center} q}$ and $\vec{b}_{reg} = [0.5\lambda, -0.5\lambda]^T$, where $q_{center}$ is the index of $\vec{H}_n$ which corresponds to the center of the maze.

Let us explain how the above neural network processes the image (see also Fig. 7). Initially $H_0$ is *On* only at the starting-point. Passing through the first convolutional-sigmoid layer it outputs $H_1$ which will be *On* (i.e. have values close to one) on all black cells which neighbor the *On* cells as well as on the original starting point. Thus *On* regions spread on the black cluster which contains the original starting-point, while white clusters and black clusters which do not contain the starting-point remain *Off* (close to zero in $H_i$). The final logistic regression layer simply checks whether one of the $2 \times 2$ cells at the center of the maze are *On* and outputs the labels accordingly.

To formalize the above we start by defining two activation thresholds, $v_l$ and $v_h$, and refer to activations which are below $v_l$ as being *Off* and to those above $v_h$ as being *On*. The quantity $v_l$ is defined as the smallest of the three real solutions of the equation $v_l = \sigma(5v_l - 0.5\lambda)$. Notably we previously chose $\lambda > \lambda_c$ as this is the critical value above which three real solutions to $v_l$ (rather than one) exist. For $v_h$ we choose 0.9.

Next, we go case by case and show that the action of the convolutional-sigmoid layer switches activations between *Off* and *On* just as a BFS would. This amounts to bounding the expression $\sigma(K_{hot} * H_{i-1} + K * I + b)$ for all possibly $3 \times 3$ sub-arrays of $H_{i-1}$ and $1 \times 1$ sub-arrays of I. There are thus $2^{10}$ possibilities to be examined.

Figure 2 shows the desired action of the layer on three important cases (A-C). Each case depicts the maze shape around some arbitrary point $x$, the hot-spot image around $x$ before the action of the layer ($H_{i-1}$), and the desired action of the layer ($H_i$). **Case A.** Having a white cell at $x$ implies $I[x] = 0$ and therefore the argument of the above sigmoid is smaller than $-0.5\lambda_c$ this regardless of $H_{i-1}$ at and around $x$. Thus $H_i[x] < v_l$ and so it is *Off*. As the 9 activations of $H_{i-1}$ played no role, case A covers in fact $2^9$ different cases. **Case B.** Consider a black cell at $x$, with $H_{i-1}$ in its vicinity all being *Off* (vicinity here refers to $x$ and its 4 neighbors). Here the argument is smaller or equal to $5v_l - 0.5\lambda_c$, and so the activation remains *Off* as desired. Case B covers $2^4$ cases as the values of $H_{i-1}$ on the 4 corners were irrelevant. **Case C.** Consider a black cell at $x$ with one or more *On* activations of $H_{i-1}$ in its vicinity. Here the argument is larger than $v_h\lambda_c - 0.5\lambda_c = 0.4\lambda_c$. The sigmoid is then larger than 0.97 implying it is *On*. Case C covers $2^4(2^5 - 1)$ different cases. Since $2^9 + 2^4 + 2^4(2^5 - 1) = 2^{10}$ we exhausted all possible cases. Lastly it can be easily verified that given an *On* (*Off*) activation at the center of the full maze the logistic regression layer will output the label *Solvable* (*Unsolvable*).

Let us now determine the required depth for this specific architecture. The previous analysis tells us that at depth $d$ unsolvable mazes would always be labelled correctly however solvable mazes would be label correctly only if the shortest-path between the starting-point and the center is $d$ or less. The worse case scenario thus occurs when the center of the maze and the starting-point are

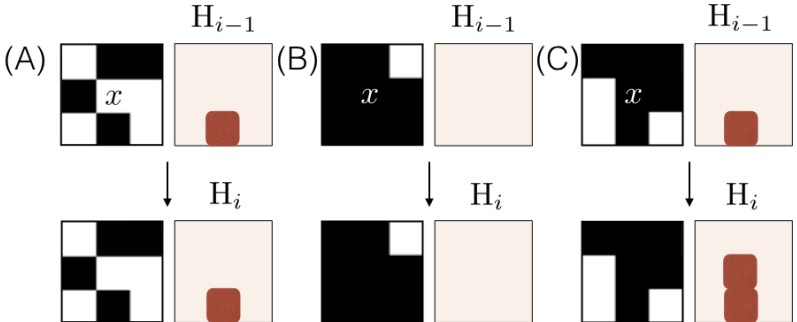

**Figure 2:** The three cases for the action of the convolutional-sigmoid layers. These cases are representative of the three sets corresponding to all possible states of the binary maze images ($I$) and the hot-spot images ($H_{i-1}$ and $H_i$), with values between 0 (*Off*, pale-orange) and 1 (*On*, red).

connected by a one dimensional curve twisting its way along $O(L^2)$ sites. Therefore, for perfect performance the network depth would have to scale as the number of sites namely $n = O(L^2)$. A tighter but probabilistic bound on the minimal depth can be established by borrowing various results from percolation theory. It is known, from Zhou et al. (2012), that the typical length of the shortest path ($l$) for critical percolation scales as $r^{d_{min}}$, where $r$ is the geometrical distance and $d_{min} = 1.1(3)$. Moreover, it is known that the probability distribution $P(l|r)$ has a tail which falls as $l^{-2}$ for $l \approx> r^{d_{min}}$ (Dokholyan et al. (1999)). Consequently, the chance that at distance $r$ the shortest path is longer than $r^{d_{min}}r^a$, where $a$ is some small positive number, decays to zero and so, $d$ should scale as $L$ with a power slightly larger than $d_{min}$ (say $n = L^{1.2}$).

## 5    TRAINING EXPERIMENTS

We have performed several training experiments with our architecture on $L = 16$ and $L = 32$ mazes with depth $n = 16$ and $n = 32$ respectively, datasets of sizes $M = 1000$, $M = 10000$, and $M = 50000$. Unless stated otherwise we used a batch size of 20 and a learning rate of 0.02. In the following, we split the experiments into two different groups corresponding to the related phenomena observed during training, which will the analyzed in detail in the next section.

**Optimal-BFS like minima.** For $L = 16$, $M = 10000$ mazes and a positive random initialization for $K_{hot}$ and $K$ in $[0, \sqrt{6/8}]$ the network found a solution with $\approx 9\%$ normalized test error performance in 3 out of the 7 different initializations (baseline test error was $50\%$). In all three successful cases the minima was a variant of the Optimal-BFS minima which we refer to as the checkerboard-BFS minima. It is similar to the optimal-BFS but spreads the *On* activations from the starting-point using a checkerboard pattern rather than a uniform one, as shown in Fig. 3. The fact that it reaches $\approx 9\%$ test error rather than zero is attributed to this checkerboard behavior which can occasionally miss out the exit point from the maze.

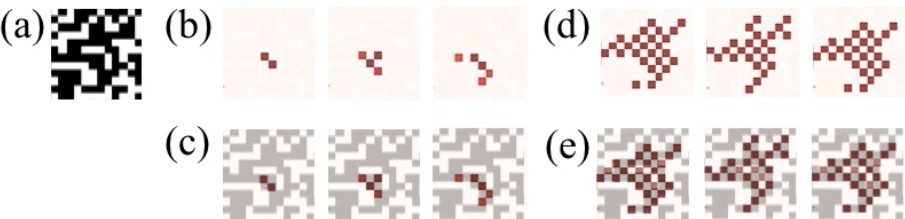

**Figure 3: Dynamics of activations for the checkerboard BFS minima obtained in training.** The activations in $H_{1..3}$ and $H_{11..13}$ are shown in (b) and (d), respectively, along with the corresponding maze (a). Superposition of the maze on top of $H_{1..3}$ and $H_{11..13}$ are shown in (c) and (e), respectively. See also a short movie with the checkerboard activations at `https://youtu.be/t-_TDkt3ER4`.

**Neglect minima.** Again for $L = 16$ but allowing for negative entries in $K$ and $K_{hot}$ test error following 14 attempts and 500 epochs did not improve below $44\%$. Analyzing the weights of the

network, the $6\%$ improvement over the baseline error ($50\%$) came solely from identifying the inverse correlation between many white cells near the center of the maze and the chance of being solvable. Notably, this heuristic approach completely neglects information regarding the starting-point of the maze. For $L = 32$ mazes, despite trying several random initialization strategies including positive entries, dataset sizes, and learning rates, the network always settled into such a partial neglect minimum. In an unsuccessful attempt to push the weights away from such partial neglect behavior, we performed further training experiments with a biased dataset in which the maze itself was uncorrelated with the label. More accurately, marginalizing over the starting-point there is an equal chance for both labels given any particular maze. To achieve this, a maze shape was chosen randomly and then many random locations were tried-out for the starting-point using that same maze. From these, we picked $5$ that resulted in a Solvable label and $5$ that resulted in an Unsolvable label. Maze shapes which were always Unsolvable were discarded. Both the $L = 16$ and $L = 32$ mazes trained on this biased dataset performed poorly and yielded $50\%$ test error. Interestingly they improved their cost function by settling into weights in which $b \approx -10$ is large compared to $[K_{hot}]_{ij} <\approx 1$ while $W$ and $\vec{b}$ were close to zero (order of $0.01$). We have verified that such weights imply that activations in the last layer have a negligible dependence on the starting-point and a weak dependence on the maze shape. We thus refer to this minimum as a "total neglect minimum".

## 6    COST FUNCTION LANDSCAPE AND THE OBSERVED TRAINING PHENOMENA

Here we seek an analytical understanding of the aforementioned training phenomena through the analysis of the cost function around solutions similar or equal to those the network settled into during training. Specifically we shall first study the cost function landscape around the optimal-BFS minimum. As would become clearer at the end of that analysis, the optimal BFS shares many similarities with the checkerboard-BFS minimum obtained during training and one thus expects a similar cost function landscape around both of these. The second phenomena analyzed below is the total neglect minimum obtained during training on the biased dataset. The total neglect minimum can be thought of as an extreme version of the partial neglect minima found for $L = 32$ in the original dataset.

### 6.1    THE SHAPE OF THE COST FUNCTION NEAR THE OPTIMAL-BFS MINIMUM

Our analysis of the cost function near the optimal-BFS minimum will be based on two separate models capturing the short and long scale behavior of the network near this miminum. In the first model we approximate the network by linearizing its action around weak activations. This model would enable us to identify the density of what we call "bugs" in the network. In the second model we discretize the activation levels of the neural network into binary variables and study how the resulting cellular automaton behaves when such bugs are introduced. Figure 4 shows a numerical example of the dynamics we wish to analyze.

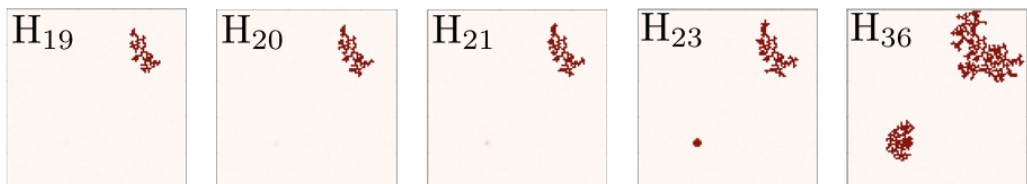

**Figure 4:** Activation dynamics for weights close to the optimal solution ($\lambda = \lambda_c - 0.227$). Up to layer 19 ($H_{19}$) the *On* activations spread according to BFS however at $H_{20}$ a very faint localized unwanted *On* activation begins to develop (a bug) and quickly saturates ($H_{23}$). Past this point BFS dynamics continues normally but spreads both the original and the unwanted *On* activations. While not shown explicitly, *On* activations still appear only on black maze cells. Notably the bug developed in rather large black region as can be deduced from the large red region in its origin. See also a short movie showing the occurrence of this bug at https://youtu.be/2I436BVAVdM and more bugs at https://youtu.be/kh-AfOo4TkU. At https://youtu.be/t-_TDkt3ER4 a similar behavior is shown for the checkerboard-BFS.

### 6.1.1 LINEARIZATION AROUND THE OPTIMAL-BFS MINIMUM AND THE EMERGENCE OF BUGS

Unlike an algorithm, a neural network is an analog entity and so a-priori there are no sharp distinctions between a functioning and a dis-functioning neural network. An algorithm can be debugged and the bug can be identified as happening at a particular time step. However it is unclear if one can generally pin-point a layer and a region within where a deep neural network clearly malfunctioned. Interestingly we show that in our toy problem such pin-pointing of errors can be done in a sharp fashion by identifying fast and local processes which cause an unwanted switching been *Off* and *On* activations in $H_i$ (see Fig. 4). We call these events bugs, as they are local, harmful, and have a sharp meaning in the algorithmic context.

Below we obtain asymptotic expressions for the chance of generating such bugs as the network weights are perturbed away from the optimal-BFS minimum. The main result of this subsection, derived below, is that the density of bugs (or chance of bug per cell) scales as

$$\rho_{bug} \propto e^{\frac{C'}{(\lambda - \lambda_c)}}, \tag{1}$$

for $(\lambda - \lambda_c) < \approx 0$ and zero for $\lambda - \lambda_c >= 0$ where $C' \approx 1.7$. Following the analysis below, we expect the same dependence to hold for generic small perturbations only with different $C'$ and $\lambda_c$. We have tested this claim numerically on several other types of perturbations (including ones that break the $\pi/2$ rotational symmetry of the weights) and found that it holds.

To derive Eq. (1), we first recall the analysis in Sec. 4, initially as it is decreased $\lambda$ has no effect but to shift $v_l$ (the *Off* activation threshold) to a higher value. However, at the critical value ($\lambda = \lambda_c, v_l = v_{l,c}$) the solution corresponding to $v_l$ vanishes (becomes complex) and the correspondence with the BFS algorithm no longer holds in general. This must not mean that all *Off* activations are no longer stable. Indeed, recall that in Sec. 4 the argument that a black *Off* cell in the vicinity of *Off* cells remains *Off* (Fig. 2, Case B) assumed a worse case scenario in which all the cells in its vicinity where both *Off*, black, and had the maximal *Off* activation allowed ($v_l$). However, if some cells in its vicinity are white, their *Off* activations levels are mainly determined by the absence of the large $K$ term in the sigmoid argument and orders of magnitude smaller than $v_l$. We come to the conclusion that black *Off* cells in the vicinity of many white cells are less prone to be spontaneously turned *On* than black *Off* cells which are part of a large cluster of black cells (see also the bug in Fig. 4). In fact using the same arguments one can show that infinitesimally below $\lambda_c$ only uniform black mazes will cause the network to malfunction.

To further quantify this, consider a maze of size $l \times l$ where the hot-spot image is initially all zero and thus *Off*. Intuitively this hot-spot image should be thought of as a sub-area of a larger maze located far away from the starting-point. In this case a functioning network must leave all activation levels below $v_l$. To assess the chance of bugs we thus study the probability that the output of the final convolutional-sigmoid layer will have one or more *On* cells.

To this end, we find it useful to linearize the system around low activation yielding (see the Appendix for a complete derivation)

$$\psi_n(r_b) = \tilde{\lambda} \left( \psi_{n-1}(r_b) + \sum_{\langle r'_b, r_b \rangle} \psi_{n-1}(r'_b) \right) + O(\tilde{\lambda}\psi_n^2), \tag{2}$$

where $r_b$ denotes black cells ($\text{I}(r_b) = 1$), the sum is over the nearest neighboring black cells to $r_b$, $\psi_n(r) = H_n(r) - v_{l,c}$, and $\tilde{\lambda} = \lambda \frac{d\sigma}{dx}(\sigma^{-1}(v_{l,c}))$.

For a given maze (I), Eq. (2), defines a linear Hermitian operator ($L_I$) with random off-diagonal matrix elements dictated by I via the restriction of the off-diagonal terms to black cells. Stability of *Off* activations is ensured if this linear operator is contracting or equivalently if all its eigenvalues are smaller than 1 in magnitude.

Hermitian operators with local noisy entries have been studied extensively in physics, in the context of disordered systems and Anderson localization (Kramer & MacKinnon (1993)). Let us describe the main relevant results. For almost all I's the spectrum of $L$ consists of localized eigenfunctions ($\phi_m$). Any such function is centered around a random site ($x_m$) and decays exponentially away from that site with a decay length of $\chi$ which in our case would be a several cells long. Thus given $\phi_m$ with an eigenvalue $|E_m| > 1$, $t$ repeated actions of the convolutional-sigmoid layer will make $\psi_n[x]$ in

a $\chi$ vicinity of $x_m$ grow in size as $e^{E_m t}$. Thus $(|E_m| - 1)^{-1}$ gives the characteristic time it takes these localized eigenvalue to grow into an unwanted localized region with an *On* activation which we define as a bug.

Our original question of determining the chance of bugs now translates into a linear algebra task: finding, $N_{\tilde{\lambda}}$, the number of eigenvalues in $L_I$ which are larger than 1 in magnitude, averaged over I, for a given $\lambda$. Since $\tilde{\lambda}$ simply scales all eigenvalues one finds that $N_{\tilde{\lambda}}$ is the number of eigenvalues larger than $\tilde{\lambda}^{-1}$ in $L_I$ with $\tilde{\lambda} = 1$. Analyzing this latter operator, it is easy to show that the maximal eigenvalues occurs when $\phi_n(r)$ has a uniform pattern on a large uniform region where the I is black. Indeed if I contains a black uniform true box of dimension $l_u \times l_u$, the maximal eigenvalue is easily shown to be $E_{l_u} = 5 - 2\pi^2/(l_u)^2$. However the chance that such a uniform region exists goes as $(l/l_u)^2 e^{\log(\rho_c)l_u^2}$ and so $P(\Delta E) \propto l^2 e^{\frac{\log(\rho_c)2\pi^2}{(\Delta E)}}$, where $\Delta E = 5 - E$. This reasoning is rigorous as far as lower bounds on $N_{\tilde{\lambda}}$ are concerned, however it turns out to capture the functional behavior of $P(\Delta E)$ near $\Delta E = 0$ accurately (Johri & Bhatt (2012)) which is given by $P(\Delta E \to 0_+) = l^2 e^{-\frac{C}{(\Delta E)}}$, where the unknown constant $C$ captures the dependence on various microscopic details. In the Appendix we find numerically that $C \approx 0.7$. Following this we find $N_{\tilde{\lambda}} \propto l^2 \int_0^{\Delta E_\lambda} dx P(x)$ where $\Delta E_\lambda = 5 - \tilde{\lambda}^{-1} \geq 0$. The range of integration is chosen to includes all eigenvalues which, following a multiplication by $\tilde{\lambda}$, would be larger than 1.

To conclude we found the number of isolated unwanted *On* activations which develop on $l \times l$ *Off* regions. Dividing this number by $l^2$ we obtain the density of bugs ($\rho_{bug}$) near $\lambda \approx \lambda_c$. The last technical step is thus to express $\rho_{bug}$ in terms of $\lambda$. Focusing on the small $\rho_{bug}$ region or $\Delta E \to 0_+$, we find that $\Delta E = 0$ occurs when $\frac{d\sigma}{dx}(\sigma^{-1}(\eta_\infty(\lambda))) = 1/(5\lambda)$, $\tilde{\lambda} = 1/5$, and $\lambda = \lambda_c = 9.72(7)$. Expanding around $\lambda = \lambda_c$ we find $\Delta E_\lambda = \frac{49 - \lambda_c}{10\lambda_c}(\lambda_c - \lambda) + O((\lambda_c - \lambda)^2)$. Approximating the integral over $P(x)$ and taking the leading scale dependence, we arrive at Eq. (1) with $C' = C\frac{10\lambda_c}{49 - \lambda_c}$.

### 6.1.2 EFFECTS OF BUGS ON BREADTH-FIRST SEARCH

In this subsection we wish to understand the large scale effect of $\rho_{bug}$ namely, its effect on the test error and the cost function. Our key results here are that

$$err \propto \rho_{bug}^{5/91} \propto e^{\frac{5C'/91}{\lambda - \lambda_c}}, \tag{3}$$

for $C'd_f^{-1}/\log(L) + O(\log^{-2}(L)) < (\lambda_c - \lambda) < C'$ where $(d_f = 91/48)$. Notably this expression is independent of L. In the domain $(\lambda_c - \lambda) <\approx C'd_f^{-1}/\log(L)$ or equivalently $\rho_{bug} <\approx L^{-d_f}$ a weak dependence on L remains and

$$err \propto L^{2 - 5/24} e^{\frac{C'}{\lambda - \lambda_c}}, \tag{4}$$

despite its appearance it can be verified that the above right hand side is smaller than $L^{-5/48}$ within its domain of applicability. Figure (5) shows a numerical verification of this last result (see Appendix for further details).

To derive Eqs. (3) and (4), we note that as a bug is created in a large maze, it quickly switches *On* the cells within the black "room" in which it was created. From this region it spreads according to BFS and turns *On* the entire cluster connected to the buggy room (see Fig. 4). To asses the effect this bug has on performance first note that solvable mazes would be labelled Solvable regardless of bugs however unsolvable mazes might appear solvable if a bug occurs on a cell which is connected to the center of the maze. Assuming we have an unsolvable maze, we thus ask what is the chance of it being classified as solvable.

Given a particular unsolvable maze instance (I), the chance of classifying it as solvable is given by $p_{err}(I) = 1 - (1 - \rho_{bug})^s = 1 - e^{-\rho_{bug}s} + O(\rho_{bug}^2)$ where $s$ counts the number of sites in the cluster connected to the central site (central cluster). The probability distribution of $s$ for percolation is known and given by $p(s) = Bs^{1-\tau}, \tau = 187/91$ (Cardy & Ziff (2003)), with $B$ being an order of one constant which depends on the underlying lattice. Since clusters have a fractional dimension, the maximal cluster size is $L^{d_f}$. Consequently, $p_{err}(I)$ averaged over all I instances is given by $p_{err} = \int_0^{L^{d_f}} p(s)[1 - e^{-\rho_{bug}s}] ds$, which can be easily expressed in terms of Gamma functions

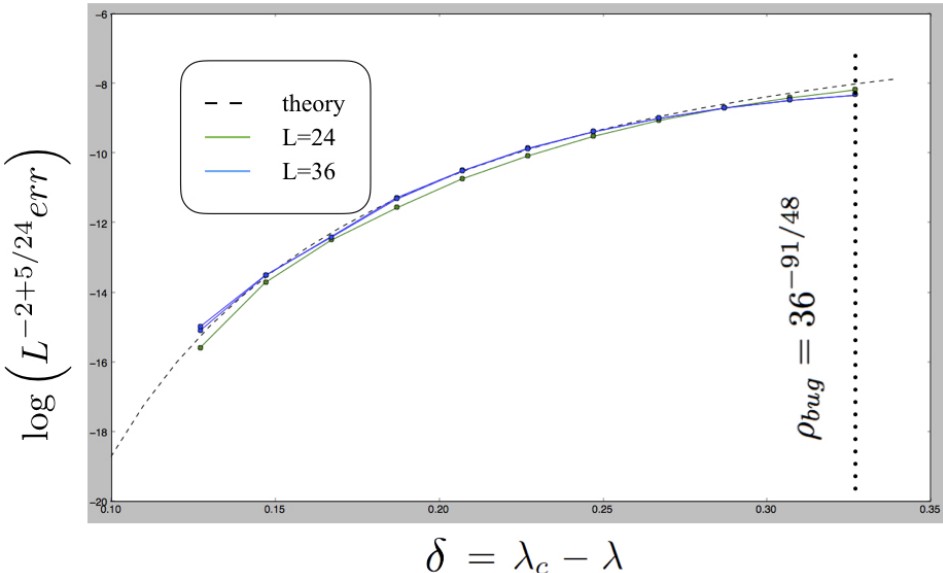

**Figure 5:** Logarithm of the numerically obtained $err$ scaled by $L^{-2+5/24}$ as a function of a deviation ($\delta$) from the optimal-BFS weights for two maze sizes along with a fit to Eq. (4). The dotted vertical line marks the end of the domain of applicability of Eq. (4).

$(\Gamma(x), \Gamma(a, x))$ (see Abramowitz (1974)). In the limit of $\rho_{bug} <\approx L^{-d_f}$, where its derivatives with respect to $\rho_{bug}$ are maximal, it simplifies to

$$p_{err} = (\tau - 2)^{-1} B\rho_{bug}L^{d_f(3-\tau)} \propto \rho_{bug}L^{2-5/24}, \tag{5}$$

whereas for $\rho_{bug} > L^{-d_f}$, its behavior changes to $p_{err} = (-B\Gamma(2-\tau))\rho_{bug}^{(\tau-2)} \propto \rho_{bug}^{5/91}$. Notably once $\rho_{bug}$ becomes of order one, several of the approximation we took break down.

Let us relate $p_{err}$ to the test error ($err$). In Sec. (2) the cost function was defined as the mislabeling chance over the average chance of being solvable ($p_{solvable}$). Following the above discussion the mislabelling chance is $p_{err}p_{solvable}$ and consequently $err = p_{err}$. Combining Eqs. 1 and 5 we obtain our key results, Eqs. (3, 4)

As a side, one should appreciate a potential training obstacle that has been avoided related to the fact that $err \propto \rho_{big}^{5/91}$. Considering $L \to \infty$, if $\rho_{bug}$ was simply proportional to $(\lambda_c - \lambda)$, $err$ will have a sharp singularity near zero. For instance, as one reduces $err$ by a factor of $1/e$, the gradients increase by $e^{86/5} \approx 3E + 7$. These effects are in accordance with ones intuition that a few bugs in a long algorithm will typically have a devastating effect on performance. Interestingly however, the essential singularity in $\rho_{bug}(\lambda)$, derived in the previous section, completely flattens the gradients near $\lambda_c$.

Thus the essentially singularity which comes directly from rare events in the dataset strongly regulates the test error and in a related way the cost function. However it also has a negative side-effect concerning the robustness of generalization. Given a finite dataset the rarity of events is bounded and so having $\lambda < \lambda_c$ may still provide perfect performance. However when encountering a larger dataset some samples with rarer events (i.e. larger black region) would appear and the network will fail sharply on these (i.e. the wrong prediction would get a high probability). Further implications of this dependence on rare events on training and generalization errors will be studied in future work.

## 6.2 COST FUNCTION NEAR A TOTAL NEGLECT MINIMA

To provide an explanation for this phenomena let us divide the activations of the upper layer to its starting-point dependent and independent parts. Let $\mathrm{H}_n$ denote the activations at the top layer. We expand them as a sum of two functions

$$\mathrm{H}_n = \alpha A(H_0, \mathrm{I}) + \beta B(\mathrm{I}) \tag{6}$$

where the function $A$ and $B$ are normalized such that their variance on the data ($\alpha$ and $\beta$, respectively) is 1. Notably near the reported total neglect minima we found that $\alpha \ll \beta \approx e^{-10}$. Also note that for the biased dataset the maze itself is uncorrelated with the labels and thus $\beta$ can be thought of as noise. Clearly any solution to the Maze testing problem requires the starting-point dependent part ($\alpha$) to become larger than the independent part ($\beta$). We argue however that in the process of increasing $\alpha$ the activations will have to go through an intermediate "noisy" region. In this noisy region $\alpha$ grows in magnitude however much less than $\beta$ and in particular obeys $\alpha < \beta^2$. As shown in the Appendix the negative log-likelihood, a commonly used cost function, is proportional to $\beta^2 - \alpha$ for $\alpha, \beta \ll 1$. Thus it penalizes random false predictions and, within a region obeying $\alpha < \beta^2$ it has a minimum (global with respect to that region) when $\alpha = \beta = 0$. The later being the definition of a total neglect minima.

Establishing the above $\alpha \ll \beta^2$ conjecture analytically requires several pathological cases to be examined and is left for future work. In this work we provide an argument for its typical correctness along with supporting numerics in the Appendix.

A deep convolution network with a finite kernel has a notion of distance and locality. For many parameters ranges it exhibits a typical correlation length ($\chi$). That is a scale beyond which two activations are statistically independent. Clearly to solve the current problem $\chi$ has to grow to an order of $L$ such that information from the input reaches the output. However as $\chi$ gradually grows, relevant and irrelevant information is being mixed and propagated onto the final layer. While $\beta$ depends on information which is locally accessible at each layer (i.e. the maze shape), $\alpha$ requires information to travel from the first layer to the last. Consequently $\alpha$ and $\beta$ are expected to scale differently, as $e^{-L/\chi}$ and $e^{-1/\chi}$ resp. (for $\chi \ll L$). Given this one finds that $\alpha \ll \beta^2$ as claimed.

Further numerical support of this conjecture is shown in the Appendix where an upper bound on the ratio $\alpha/\beta^2$ is studied on 100 different paths leading from the total neglect miminum found during training to the checkerboard-BFS minimum. In all cases there is a large region around the total neglect minimum in which $\alpha \ll \beta^2$.

## 7 CONCLUSIONS

Despite their black-box reputation, in this work we were able to shed some light on how a particular deep CNN architecture learns to classify topological properties of graph structured data. Instead of focusing our attention on general graphs, which would correspond to data in non-Euclidean spaces, we restricted ourselves to planar graphs over regular lattices, which are still capable of modelling real world problems while being suitable to CNN architectures.

We described a toy problem of this type (Maze-testing) and showed that a simple CNN architecture can express an exact solution to this problem. Our main contribution was an asymptotic analysis of the cost function landscape near two types of minima which the network typically settles into: BFS type minima which effectively executes a breadth-first search algorithm and poorly performing minima in which important features of the input are neglected.

Quite surprisingly, we found that near the BFS type minima gradients do not scale with $L$, the maze size. This implies that global optimization approaches can find such minima in an average time that does not increase with $L$. Such very moderate gradients are the result of an essential singularity in the cost function around the exact solution. This singularity in turn arises from rare statistical events in the data which act as early precursors to failure of the neural network thereby preventing a sharp and abrupt increase in the cost function.

In addition we identified an obstacle to learning whose severity scales with $L$ which we called neglect minima. These are poorly performing minima in which the network neglects some important features relevant for predicting the label. We conjectured that these occur since the gradual incorporation of these important features in the prediction requires some period in the training process in which predictions become more noisy. A "wall of noise" then keeps the network in a poorly performing state.

It would be interesting to study how well the results and lessons learned here generalize to other tasks which require very deep architectures. These include the importance of rare-events, the essential

singularities in the cost function, the localized nature of malfunctions (bugs), and neglect minima stabilized by walls of noise.

These conjectures potentially could be tested analytically, using other toy models as well as on real world problems, such as basic graph algorithms (e.g. shortest-path) (Graves et al. (2016)); textual reasoning on the bAbI dataset (Weston et al. (2015)), which can be modelled as a graph; and primitive operations in "memory" architectures (e.g. copy and sorting) (Graves et al. (2014)). More specifically the importance of rare-events can be analyzed by studying the statistics of errors on the dataset as it is perturbed away from a numerically obtained minimum. Technically one should test whether the perturbation induces an typical small deviation of the prediction on most samples in the dataset or rather a strong deviation on just a few samples. Bugs can be similarly identified by comparing the activations of the network on the numerically obtained minimum and on some small perturbation to that minimum while again looking at typical versus extreme deviations. Such an analysis can potentially lead to safer and more robust designs were the network fails typically and mildly rather than rarely and strongly.

Turning to partial neglect minima these can be identified provided one has some prior knowledge on the relevant features in the dataset. The correlations or mutual information between these features and the activations at the final layer can then be studied to detect any sign of neglect. If problems involving neglect are discovered it may be beneficial to add extra terms to the cost function which encourage more mutual information between these neglected features and the labels thereby overcoming the noise barrier and pushing the training dynamics away from such neglect minimum.

## ACKNOWLEDGMENTS

Rodrigo Andrade de Bem is a CAPES Foundation scholarship holder (Process no: 99999.013296/2013-02, Ministry of Education, Brazil).

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

## A    VISUALIZATION OF THE OPTIMAL-BFS MINIMUM

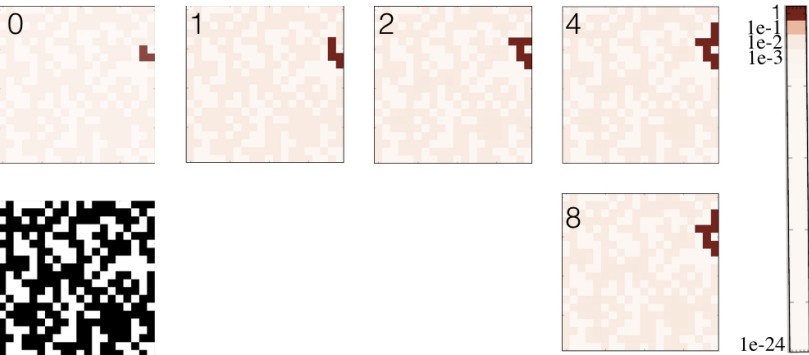

**Figure 6:** A numerical experiment showing how our maze classification architecture processes a particular sample consisting a maze (black and white image) and a hot-spot image marking the starting-point (panel (0)) when its weights are tuned to the optimal BFS solution. The first layer receives a hot-spot image which is *On* only near the starting-point of the maze $H_0$) (panel (0)). This *On* activation then spreads on the black cluster containing the start-point in ($H_n$ with $n = 1, 2, 4, 8$, panels 1,2,4,8 resp.). Notably other region are *Off* (i.e. smaller than $v_l$) but they are not zero as shown by the faint imprint of the maze on $H_n$.

## B   TEST ERROR AWAY FROM THE OPTIMAL-BFS SOLUTION

We have implemented the architecture described in the main text using Theano (Theano Development Team (2016)) and tested how *cost* changes as a function of $\delta = \lambda_c - \lambda$ ($\lambda_c = 9.727..$) for mazes of sizes $L = 24, 36$ and depth (number of layers) 128. These depths are enough to keep the error rate negligible at $\delta = 0$. A slight change made compared to Maze-testing as described in the main text, is that the hot-spot was fixed at a distance $L/2$ for all mazes. The size of the datasets was between $1E + 5$ and $1E + 6$. We numerically obtained the normalized performance ($cost_L(\delta)$) as a function of $L$ and $\delta$.

As it follows from Eq. (4) in the main text the curve, $\log(L^{-2+5/24}cost_L(\delta))$, for the $L = 24$ and $L = 36$ results should collapse on each other for $\rho_{bug} < L^{-d_f}$. Figure (5) of the main-test depicts three such curves, two for $L = 36$, to give an impression of statistical error, and one for $L = 24$ curve (green), along with the fit to the theory (dashed line). The fit, which involves two parameters (the proportionally constant in Eq. (4) of the main text and $C$) captures well the behavior over three orders of magnitude. As our results are only asymptotic, both in the sense of large $L$ and $\lambda \to \lambda_c$, minor discrepancies are expected.

## C   LINEARIZATION OF THE SIGMOID-CONVOLUTIONAL NETWORK AROUND OFF ACTIVATION

To prepare the action of sigmoid-convolutional for linearization we find it useful to introduce the following variables on locations ($r_b, r_w$) with black ($b$) and white ($w$) cells

$$\psi_n(r_\alpha) = H_n(r_\alpha) - a(r_\alpha) \tag{7}$$
$$a(r_b) = v_{l,c} \tag{8}$$
$$a(r_w) = e^{-5.5\lambda}. \tag{9}$$

Rewriting the action of the sigmoid-convolutional layer in terms of these we obtain

$$\psi_n(r_\alpha) + a(r_\alpha) = \sigma \left[ \lambda \left( \psi_{n-1}(r_\alpha) + \sum_{\langle r',r \rangle} \psi_{n-1}(r') + 5I(r_\alpha) \right) - 5.5\lambda + \lambda d(r_\alpha) \right], \quad (10)$$

$$d(r_\alpha) = a(r_\alpha) + \sum_{\langle r',r_\alpha \rangle} a(r')$$

where $\sum_{\langle r',r \rangle}$ means summing over the 4 sites neighboring $r$. Next we treating $\psi_n(r)$ as small and Taylor expand

$$\psi_n(r_w) = \lambda \frac{\sigma}{dx}|_{-5.5\lambda} \left( \psi_{n-1}(r_\alpha) + \sum_{\langle r',r_w \rangle} \psi_{n-1}(r') + d(r_w) \right) \quad (11)$$

$$\psi_n(r_b) = \lambda \frac{d\sigma}{dx}|_{\sigma^{-1}(v_{l,c})} \left( \psi_{n-1}(r_\alpha) + \sum_{\langle r',r_b \rangle} \psi_{n-1}(r') + (d(r_b) - 5v_{l,c}) \right)$$

where $v_{l,c} \approx 0.02(0)$ is the low (and marginally stable) solution of the equation $v_{l,c} = \sigma(-0.5\lambda_c + 5v_{l,c})$.

Next in the consistency with our assumption that $|\psi_{n-1}(r)|$ is small, we can assume $|\psi_{n-1}(r)| < 1$, and obtain that $\psi_n(r_w) < e^{-5.5\lambda}(5 + e^{-0.5\lambda})$ and therefore, since we are working near $\lambda = 9.727..$ it is negligible. The equation of $\psi_n(r_b)$ now appears as

$$\psi_n(r_b) = \tilde{\lambda} \left( \psi_{n-1}(r_\alpha) + \sum_{\langle r'_b,r_b \rangle} \psi_{n-1}(r') + d(r_b) - 5v_{l,c} \right) + O(\lambda \frac{d^2\sigma}{dx^2}\psi^2) \quad (12)$$

where the summation of neighbor now includes only black cells and $\tilde{\lambda} = \lambda \frac{d\sigma}{dx}|_{\sigma^{-1}(v_{l,c})}$. Due to form the sigmoid function, $\lambda \frac{d^2\sigma}{dx^2}|_{\sigma^{-1}[\epsilon_\infty]}$ is of the same magnitude as $\tilde{\lambda}$, and consequently the relative smallness of this terms is guaranteed as long as $\psi_n \ll 1$.

We thus obtained a linearized version for the sigmoid-convolutional network which is suitable for stability analysis. Packing $\psi_n(r_b)$ and $d(r_b) - 5v_{l,c}$ into vectors $(\vec{\psi}_n, \vec{d(r_b)})$ the equation we obtained can be written as

$$\vec{\psi}_n = S\vec{\psi}_{n-1} + \vec{d} \quad (13)$$

with $S$ being a symmetric matrix. Denoting by $\vec{\phi}_n^T$ and $s_n$ the left eigenvectors and eigenvalues of $S$, we multiply the above equation from the left with $\vec{\phi}_n^T$ and obtain

$$c_n = s_n c_{n-1} + \vec{\phi}_n^T \vec{d} \quad (14)$$

$$\vec{\psi}_n = \sum_n c_n \vec{\phi}_n.$$

Stability analysis on this last equation is straightforward: For $|s_n| < 1$, a stable solution exists given by $c_n = \frac{\vec{\phi}_n^T \vec{d}}{(1-s_n)}$. Furthermore as the matrix $S$ has strong disorder, $\vec{\phi}_n$ are localized in space. Consequently $\vec{\phi}_n^T \vec{d}$ is of the same order of magnitude as $\vec{d} \approx e^{-0.5\lambda} \approx 0.01$ and as long as $s_n < 0.9$, these stable solutions are well within the linear approximation we have carried. For $|s_n| > 1$, there are no stable solutions.

There is an important qualitative lesson to be learned from applying these results on an important test case: A maze with only black cells. In this case it is easy to verify directly on the non-linear sigmoid-convolutional map that a uniform solution becomes unstable exactly at $\lambda = \lambda_c$. Would we find the same result within our linear approximation?

To answer the above, first note that the maximal eigenvalue of $S$ will be uniform with $s_{max} = 5\tilde{\lambda}$. Furthermore for an all black maze $\vec{d}$ would be exactly zero and the linear equation becomes

homogeneous. Consequently destabilization occurs exactly at $\tilde{\lambda} = 1/5$ and is not blurred by the inhomogeneous terms. Recall that $\lambda_c$ is defined as the value at which the two lower solutions of $x = \sigma[-0.5\lambda_c + 5\lambda_c x]$ and it also satisfies the equation $v_{l,c} = \sigma[-0.5\lambda_c + 5\lambda_c v_{l,c}]$. Taking a derivative of the former and putting $x = v_{l,c}$ one finds that $1 = 5\lambda_c \frac{d\sigma[-0.5\lambda_c + 5v_{l,c}]}{dx}$. It is now easy to verify that even within the linear approximation destabilization occurs exactly at $\lambda_c$. The source of this agreement is the fact that $\vec{d}$ vanishes for a uniform black maze.

The qualitative lesson here is thus the following: The eigenvectors of $S$ with large $s$, are associated with large black regions in the maze. It is only on the boundaries of such regions that $\vec{d}$ is non-zero. Consequently near $\lambda \approx \lambda_c$ the $\vec{d}$ term projected on the largest eigenvalues can, to a good accuracy, be ignored and stability analysis can be carried on the homogeneous equation $\vec{\psi} = S\vec{\psi}$ where $s_n < 1$ means stability and $s_n > 1$ implies a bug.

# D   Log-likelihood and noisy predictions

Consider an abstract classification tasks where data point $x \in X$ are classified into two categories $l \in \{0, 1\}$ using a deterministic function $f : X \to \{0, 1\}$ and further assume for simplicity that the chance of $f(x) = a$ is equal to $f(x) = b$. Phrased as a conditional probability distribution $P_f(l|x)$ is given by $P_f(f(a)|x) = 1$ while $P_f(!f(a)|x) = 0$. Next we wish to compare the following family of approximations to $P_f$

$$P_{\alpha',\beta'}(l|x) = 1/2 + \alpha'(2l-1)(2f(x)-1) + \beta'(2l-1)(2g(x)-1) \tag{15}$$

where $g|X \to \{0, 1\}$ is a random function, uncorrelated with $f(x)$, outputting the labels $\{0, 1\}$ with equal probability. Notably at $\alpha' = 1/2, \beta' = 0$ it yields $P_f$ while at $\alpha', \beta' = 0$ it is simply the maximum entropy distribution.

Let us measure the log-likelihood of $P_{\alpha',\beta'}$ under $P_f$ for $\alpha', \beta' \ll 1$

$$\mathcal{L}(\alpha', \beta') = \sum_{(x,l)} P_f(x,l) \log\left(1/2 + \alpha'(2l-1)(2f(x)-1) + \beta'(2l-1)(2g(x)-1)\right) \tag{16}$$

$$\approx \sum_{(x,l)} P_f(x,l) \log(1/2) + 2\left[\alpha'(2l-1)(2f(x)-1) + \beta'(2l-1)(2g(x)-1)\right]$$

$$- 2\left[\alpha'(2l-1)(2f(x)-1) + \beta'(2l-1)(2g(x)-1)\right]^2$$

$$= \log(1/2) + 2\alpha' - 2\alpha'^2 - 2\beta'^2$$

We thus find that $\beta'$ reduces the log-likelihood in what can be viewed as a penalty to false confidence or noise. Assuming, as argued in the main text, that $\alpha'$ is constrained to be smaller than $\beta'^2$ near $\beta' \approx 0$, it is preferable to take both $\alpha'$ and $\beta'$ to zero and reach the maximal entropy distribution. We note by passing that the same arguments could be easily generalized to $f(x), g(x)$ taking real values leading again to an $O(\alpha) - O(\beta^2)$ dependence in the cost function.

Let us relate the above notations to the ones in the main text. Clearly $x = (\{I\}, H_0)$ and $\{0, 1\} = \{Unsolvable, Solvable\}$. Next we recall that in the main text $\alpha$ and $\beta$ multiplied the vectors function representing the $H_0$-depended and $H_0$-independent parts of $H_n$. The probability estimated by the logistic regression module was given by

$$P(Solvable|x) = \frac{e^{\vec{K}_{Solvable} \cdot \vec{H}_n}}{e^{-\vec{K}_{Solvable} \cdot \vec{H}_n} + e^{-\vec{K}_{Unsolvable} \cdot \vec{H}_n}} \tag{17}$$

$$P(Unsolvable|x) = \frac{e^{\vec{K}_{Unsolvable} \cdot \vec{H}_n}}{e^{-\vec{K}_{Solvable} \cdot \vec{H}_n} + e^{-\vec{K}_{Unsolvable} \cdot \vec{H}_n}}$$

which yields, to leading order in $\alpha$ and $\beta$

$$P_{\alpha,\beta}(l|x) = 1/2 + \alpha(2l+1)\vec{K}_l^- \cdot A + \beta(2l+1)\vec{K}_l^- \cdot B \tag{18}$$

where $\vec{K}^- = (\vec{K}_{Solvable} - \vec{K}_{Unsolvable})/2$ and $(2l+1)$ understood as the taking the values $\pm 1$. Consequently $(2f - 1)$ and $(2g - 1)$ are naturally identified with $\vec{K}_{Solvable} \cdot A/N_A$ and $\vec{K}_{Solvable} \cdot B/N_B$ respectively with $N_A$ and $N_B$ being normalization constants ensuring a variance of 1. While $(\alpha', \beta') = (N_A\alpha, N_B\beta)$. Recall also that by construction of the dataset, the $g$ we thus obtain is uncorrelated with $f$.

# E    NUMERICAL SUPPORT FOR THE $\alpha \ll \beta^2$ CONJECTURE

Here we provide numerical evidence showing that $\alpha \ll \beta^2$ in a large region around the total neglect minima found during the training of our architecture on the biased dataset (i.e. the one where marginalizing over the starting-point yields a 50/50 chance of being solvable regardless of the maze shape).

For a given set of $K_h ot, K$ and $b$ parameters we fix the maze shape and study the variance of the top layer activations given $O(100)$ different starting points. We pick the maximal of these and then average this maximal variance over $O(100)$ different mazes. This yields our estimate of $\alpha$. In fact it is is an upper bound on $\alpha$ as this averaged-max-variance may reflect wrong prediction provided that they depend on $H_0$.

We then obtain an estimate of $\beta$ by again calculating the average-max-variance of the top layer however now with $H_0 = 0$ for all maze shapes.

Next we chose a 100 random paths parametrized by $\gamma$ leading from the total neglect minima ($\gamma = 0$) for the total neglect through a random point at $\gamma = 15$, and then to the checkerboard-BFS minima at $\gamma = 30$. The random point was placed within a hyper-cube of length 4 having the total neglect minima at its center. The path was a simple quadratic interpolation between the three point. The graph below shows the statistics of $\alpha/\beta^2$ on these 100 different paths. Notably no path even had $\alpha > e^{-30}\beta^2$ within the hyper-cube. We have tried three different other lengths for the hyper cube (12 and 1) and arrived at the same conclusions.

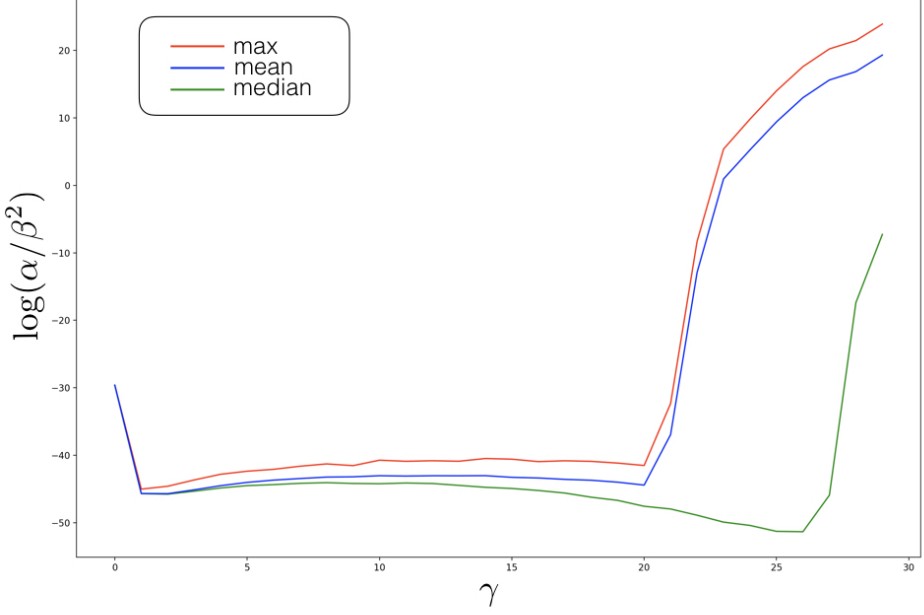

**Figure 7:** The natural logarithm of an upper bound to $\alpha/\beta^2$ as a function of a paramterization ($\gamma$) of a path leading from the numerically obtained total neglect minima to the checkerboard BFS minima through a random point. The three different curves show the max,mean, and median based on a 100 different paths. Notably no path violated the $\alpha \ll \beta^2$ constrain in the vicinity of the total neglect minima.

