# OpenReview forum: "Critical Percolation as a Framework to Analyze the Training of Deep Networks"
_ICLR.cc/2018/Conference — Accept (Poster)_

### Official Review · AnonReviewer2 · 2017-11-27
**Interesting theoretical work on CNNs via toy example**

**Rating:** 7
**Confidence:** 3

**Review:**

The authors are motivated by two problems: Inputting non-Euclidean data (such as graphs) into deep CNNs, and analyzing optimization properties of deep networks. In particular, they look at the problem of maze testing, where, given a grid of black and white pixels, the goal is to answer whether there is a path from a designated starting point to an ending point.

They choose to analyze mazes because they have many nice statistical properties from percolation theory. For one, the problem is solvable with breadth first search in O(L^2) time, for an L x L maze. They show that a CNN can essentially encode a BFS, so theoretically a CNN should be able to solve the problem. Their architecture is a deep feedforward network where each layer takes as input two images: one corresponding to the original maze (a skip connection), and the output of the previous layer. Layers alternate between convolutional and sigmoidal. The authors discuss how this architecture can solve the problem exactly. The pictorial explanation for how the CNN can mimic BFS is interesting but I got a little lost in the 3 cases on page 4. For example, what is r? And what is the relation of the black/white and orange squares? I thought this could use a little more clarity.

Though experiments, they show that there are two kinds of minima, depending on whether we allow negative initializations in the convolution kernels. When positive initializations are enforced, the network can more or less mimic the BFS behavior, but never when initializations can be negative. They offer a rigorous analysis into the behavior of optimization in each of these cases, concluding that there is an essential singularity in the cost function around the exact solution, yet learning succumbs to poor optima due to poor initial predictions in training.

I thought this was an impressive paper that looked at theoretical properties of CNNs. The problem was very well-motivated, and the analysis was sharp and offered interesting insights into the problem of maze solving. What I thought was especially interesting is how their analysis can be extended to other graph problems; while their analysis was specific to the problem of maze solving, they offer an approach -- e.g. that of finding "bugs" when dealing with graph objects -- that can extend to other problems. I would be excited to see similar analysis of other toy problems involving graphs.

One complaint I had was inconsistent clarity: while a lot was well-motivated and straightforward to understand, I got lost in some of the details (as an example, the figure on page 4 did not initially make much sense to me). Also, in the experiments, the authors mention multiple attempt with the same settings -- are these experiments differentiated only by their initialization? Finally, there were various typos throughout (one example is "neglect minimua" on page 2 should be "neglect minima").

Pros: Rigorous analysis, well motivated problem, generalizable results to deep learning theory
Cons: Clarity

---

> ### Author Response · Authors · 2017-12-12
> **Author's response to AnonReviewer2**
>
> AUTHORS: We would first like to thank the reviewer his their valuable comments and feedback. We were glad to hear that all reviewers agreed that our paper has novel and positive points. We strived to follow the reviewers’ comments to improve it, especially increasing its clarity. The changes can be viewed in the revised version already submitted. Here we address all the concerns raised by the reviewers, pointing out the correspondent modifications made in the text (all references made to section, figures and equation are w.r.t. their number in the revised version of the paper):
>
> MAIN COMMENTS:
>
> 1) The pictorial explanation for how the CNN can mimic BFS is interesting but I got a little lost in the 3 cases on page 4. For example, what is r? And what is the relation of the black/white and orange squares? I thought this could use a little more clarity.
>
> AUTHORS: First we'd like to the comment that the revised version contains many new illustrations, numerical results, and movies which are meant to improve the clarity of our presentation. More specifically to the above comment  we have improved the explanation in Section 4, correcting nomenclature (referring to x instead of r to denote a position in the maze or in the hot-spot images) and improved Figure 2 where the 3 cases in question are illustrated. Already in Figure 1 we added the explanation: “A maze-testing sample consists of a maze (I) and an initial hot-spot image (H0). The proposed architecture processes H0 by generating a series of hot-spot images (Hi>0) which are of the same dimension as H0, however their pixels are not binary but rather take on values between 0 (Off, pale-orange) and 1 (On, red).” We have added a similar explanation in the legend of Figure 2.
>
> 2) One complaint I had was inconsistent clarity: while a lot was well-motivated and straightforward to understand, I got lost in some of the details (as an example, the figure on page 4 did not initially make much sense to me).
>
> AUTHORS: As mentioned previously, we have improved the clarity of Section 4, along with other parts of text.
>
> 3) Also, in the experiments, the authors mention multiple attempt with the same settings -- are these experiments differentiated only by their initialization?
>
> AUTHORS: Yes. This has now been made clearer in the text.
>
> 4) Finally, there were various typos throughout (one example is "neglect minimua" on page 2 should be "neglect minima").
>
> AUTHORS: We have corrected these and additional typos in the text.
>
>
> MAIN SUGGESTIONS:
>
> I would be excited to see similar analysis of other toy problems involving graphs.
>
> AUTHORS: We appreciate this suggestion which was added as possible future works to our conclusion section.

---

### Official Review · AnonReviewer3 · 2017-11-27
**Good contribution, paper needs to be made clearer**

**Rating:** 7
**Confidence:** 3

**Review:**

This paper thoroughly analyzes an algorithmic task (determining if two points in a maze are connected, which requires BFS to solve) by constructing an explicit ConvNet solution and analytically deriving properties of the loss surface around this analytical solution. They show that their analytical solution implements a form of BFS algorithm, characterize the probability of introducing "bugs" in the algorithm as the weights move away from the optimal solution, and how this influences the error surface for different depths. This analysis is conducted by drawing on results from the field of critical percolation in physics.

Overall, I think this is a good paper and its core contribution is definitely valuable: it provides a novel analysis of an algorithmic task which sheds light on how and when the network fails to learn the algorithm, and in particular the role which initialization plays. The analysis is very thorough and the methods described may find use in analyzing other tasks. In particular, this could be a first step towards better understanding the optimization landscape of memory-augmented neural networks (Memory Networks, Neural Turing Machines, etc) which try to learn reasoning tasks or algorithms. It is well-known that these are sensitive to initialization and often require running the optimizer with multiple random seeds and picking the best one. This work actually explains the role of initialization for learning BFS and how certain types of initialization lead to poor solutions. I am curious if a similar analysis could be applied to methods evaluated on the bAbI question-answering tasks (which can be represented as graphs, like the maze task) and possibly yield better initialization or optimization schemes that would remove the need for multiple random seeds.

With that being said, there is some work that needs to be done to make the paper clearer. In particular, many parts are quite technical and may not be accessible to a broader machine learning audience. It would be good if the authors spent more time developing intuition (through visualization for example) and move some of the more technical proofs to the appendix. Specifically:
- I think Figure 3 in the appendix should be moved to the main text, to help understand the behavior of the analytical solution.
- Top of page 5, when you describe the checkerboard BFS: please include a visualization somewhere, it could be in the Appendix.
- Section 6: there is lots of math here, but the main results don't obviously stand out. I would suggest highlighting equations 2 and 4 in some way (for example, proposition/lemma + proof), so that the casual reader can quickly see what the main results are. Interested readers can then work through the math if they want to. Also, some plots/visualizations of the loss surface given in Equations 4 and 5 would be very helpful.

Also, although I found their work to be interesting after finishing the paper, I was initially confused by how the authors frame their work and where the paper was heading. They claim their contribution is in the analysis of loss surfaces (true) and neural nets applied to graph-structured inputs. This second part was confusing - although the maze can be viewed as a graph, many other works apply ConvNets to maze environments [1, 2, 3], and their work has little relation to other work on graph CNNs. Here the assumptions of locality and stationarity underlying CNNs are sensible and I don't think the first paragraph in Section 3 justifying the use of the CNN on the maze environment is necessary. However, I think it would make much more sense to mention how their work relates to other neural network architectures which learn algorithms (such as the Neural Turing Machine and variants) or reasoning tasks more generally (for example, memory-augmented networks applied to the bAbI tasks).

There are lots of small typos, please fix them. Here are a few:
- "For L=16, batch size of 20, ...": not a complete sentence.
- Right before 6.1.1: "when the these such" -> "when such"
- Top of page 8: "it also have a" -> "it also has a", "when encountering larger dataset" -> "...datasets"
-  First sentence of 6.2: "we turn to the discuss a second" -> "we turn to the discussion of a second"
- etc.

Quality: High
Clarity: medium-low
Originality: high
Significance: medium-high

References:
[1] https://arxiv.org/pdf/1602.02867.pdf
[2] https://arxiv.org/pdf/1612.08810.pdf
[3] https://arxiv.org/pdf/1707.03497.pdf

---

> ### Author Response · Authors · 2017-12-12
> **Author's reply to AnonReviewer 3**
>
> AUTHORS: We would first like to thank the reviewer his their valuable comments and feedback. We were glad to hear that all reviewers agreed that our paper has novel and positive points. We strived to follow the reviewers’ comments to improve it, especially increasing its clarity. The changes can be viewed in the revised version already submitted. Here we address all the concerns raised by the reviewers, pointing out the correspondent modifications made in the text (all references made to section, figures and equation are w.r.t. their number in the revised version of the paper):
>
> MAIN COMMENTS:
>
> 1) In particular, many parts are quite technical and may not be accessible to a broader machine learning audience. It would be good if the authors spent more time developing intuition.... Specifically:
>
> - I think Figure 3 in the appendix should be moved to the main text, to help understand the behaviour of the analytical solution.
>
> AUTHORS: We have merged former Figure 3 (Appendix) with former Figure 1 (main text), in what is now the current Figure 1 (main text). We belief this figure now illustrate better the whole architecture as well as how each sample of the toy dataset is composed. In the legend of the current Figure 1 we have added a lot more details about the dataset, the architecture and the breadth-first search optimum.
>
> - Top of page 5, when you describe the checkerboard BFS: please include a visualization somewhere, it could be in the Appendix.
>
> AUTHORS: We have added the Figure 3, in which we included a visualization of the checkerboard BFS pattern. Additionally we have included Figure 4, which illustrate the occurrence of bugs in the process. We also have included links to videos showing the layer by layer activation levels for these two phenomena.
>
> - Section 6: there is lots of math here, but the main results don't obviously stand out. I would suggest highlighting equations 2 and 4 in some way (for example, proposition/lemma + proof)..... Also, some plots/visualizations of the loss surface given in Equations 4 and 5 would be very helpful.
>
> AUTHORS: As suggested, we have made section 6 more concise, highlighting and simplifying former equations 2, 4 and 5 (current equations 1, 3 and 4). These key results were moved to the beginning of their subsections and are then followed by their derivations. We hope this simplifies the reading as required. Regarding visualization of the loss function, this now appears in Fig. 5 which plots the logarithm of the error near the optimal BFS solution.
>
> 2) I was initially confused by how the authors frame their work and where the paper was heading. They claim their contribution is in the analysis of loss surfaces (true) and neural nets applied to graph-structured inputs.
> This second part was confusing - although the maze can be viewed as a graph, many other works apply ConvNets ...
> Here the assumptions of locality and stationarity underlying CNNs are sensible and I don't think the first paragraph in Section 3 justifying the use of the CNN on the maze environment is necessary.
> However, I think it would make much more sense to mention how their work relates to other neural network architectures which learn algorithms ...
>
> AUTHORS: We appreciated the above comments from R3. We included the cited references, from the realm of reinforced learning and planning, into our introduction in the context of related works (fourth paragraph). We also simplified the first paragraph in Section 3 suppressing unnecessary justifications.
> Concerning the modelling of mazes as graphs, although we acknowledge that [1,2,3] do not directly or strongly relate mazes to graphs, this approach is commonly adopted in graph theory and applications. We also referred to additional works in the introduction to support our claim (fourth paragraph). Moreover, in doing so, we belief we were capable of: i) establishing a correspondence between our network and the well-know BFS algorithm; ii) pointing out real problems in which planar graph can be used to model the data; iii) setting up the introduced analysis as the basis for an eventual extension of it to general graphs.
> Finally, we also have added references and explanations based on the insightful comment about the relation between our framework and “memory” networks. These points were added in the penultimate paragraph of the introduction.
>
> 3) There are lots of small typos, please fix them....
>
> AUTHORS: We have corrected all the aforementioned typos and some additional ones.
>
> MAIN SUGGESTIONS:
>
> I am curious if a similar analysis could be applied to methods evaluated on the bAbI question-answering tasks (which can be represented as graphs, like the maze task) and possibly yield better initialization or optimization schemes that would remove the need for multiple random seeds.
>
> AUTHORS: We appreciate this suggestion which was added as possible future works to our conclusion section.

---

### Official Review · AnonReviewer1 · 2017-11-30
**novel but difficult to read and hard to assess.**

**Rating:** 6
**Confidence:** 1

**Review:**

This paper studies a toy problem: a random binary image is generated, and treated as a maze (1=wall, 0=freely moveable space). A random starting point is generated. The task is to learn whether the center pixel is reachable from the starting point.

A deep architechture is proposed to solve the problem: see fig 1. A conv net on the image is combined with that on a state image, the state being interpreted as rechable pixels. This can work if each layer expands the reachable region (the state) by one pixel if the pixel is not blocked.

Two local minima are observed: 1) the network ignores stucture and guesses if the task is solvable by aggregate statistics 2) it works as described above but propagates the rechable region on a checkerboard only.

The paper is chiefly concerned with analysing these local minima by expanding the cost function about them. This analysis is hard to follow for non experts graph theory. This is partly because many non-trivial results are mentioned with little or no explanation.

The paper is hard to evaluate. The actual setup seems somewhat arbitrary, but the method of analysing the failure modes is interesting. It may inspire more useful research in the future.

If we trust the authors, then the paper seems good because it is fairly unusual. But it is hard to determine whether the analysis is correct.

---

> ### Author Response · Authors · 2017-12-12
> **Author's reply to AnonReviewer1**
>
> AUTHORS: We would first like to thank the reviewer his their valuable comments and feedback. We were glad to hear that all reviewers agreed that our paper has novel and positive points. We strived to follow the reviewers’ comments to improve it, especially increasing its clarity. The changes can be viewed in the revised version already submitted. Here we address all the concerns raised by the reviewers, pointing out the correspondent modifications made in the text (all references made to section, figures and equation are w.r.t. their number in the revised version of the paper):
>
> MAIN COMMENTS:
> 1) This analysis is hard to follow for non experts in graph theory. This is partly because many non-trivial results are mentioned with little or no explanation.
>
> AUTHORS: Following the referee’s comment we improved the text along the following three main lines:
> 1. Further numerical results in the form of snapshots of activations levels and movies tracking the development of activation levels as a function of the layers, were added to the work. This provides direct evidence supporting our analytical claims.
> 2. The key results in our analytical analysis carried in section 6. have been brought forward so that the key results of the analytical derivation is clearer.
> 3. The numerical verification of our key results have been brought from the appendix into the main text (Fig 5.)
> We hope that this would increase the referee’s confidence in our claims.
>
> 2) The paper is hard to evaluate. The actual setup seems somewhat arbitrary, but the method of analysing the failure modes is interesting.
>
> AUTHORS: The reason we have chosen the proposed toy model is two-fold: i) it allows the modelling of data structured as planar graphs, an important and actual research topic in the field;
> ii) it allows the use of various tools from theoretical physics to derive non-trivial analytical results from the data. Indeed believe that our analysis of failure mode will be relevant for other problems as well.
>
> 3) If we trust the authors, then the paper seems good because it is fairly unusual. But it is hard to determine whether the analysis is correct.
>
> AUTHORS: To better support the correctness of the presented analysis, we have enhanced the main text with more experimental results and illustrations in the form of figures and movies. Some of them were previously placed in the Appendix (Fig. 5), while others were added to the text (Fig. 3 and 4).  We also mention that in Fig. 5 our theoretical predictions regarding the error landscape are shown to compare well with numerical experiments.

---

### Decision · Program_Chairs · 2018-01-29
**ICLR 2018 Conference Acceptance Decision**

**Decision:**

Accept (Poster)

**Comment:**

The paper got generally positive scores of 6,7,7. The reviewers found the paper to be novel but hard to understand. The AC feels the paper should be accepted but the authors should revise their paper to take into account the comments from the reviewers to improve clarity.